

# Teaching Uncertainty: A new framework for communicating unknowns in traditional and virtual field experiences

Cristina G. Wilson[1,2], Randolph T. Williams[3], Kathryn Bateman[2], Basil Tikoff[3], Thomas F. Shipley[2]

[1]General Robotics, Automation, Sensing, and Perception Laboratory, School of Engineering and Applied Science, University of Pennsylvania, Philadelphia, PA, USA
[2]Department of Psychology, College of Arts and Sciences, Temple University, Philadelphia, PA, USA
[3]Department of Geoscience, College of Letters and Science, University of Wisconsin-Madison, Madison, WI, USA

*Correspondence to*: Cristina G. Wilson (wilsoncg@seas.upenn.edu)

**Abstract.** Managing uncertainty is fundamental to geoscience practice, yet geoscience education generally does not incorporate explicit instruction on uncertainty. To the extent that students are exposed to scientific uncertainty, it is through in-person field experiences. Virtual field experiences – which rely on pictures, maps, and previously collected measurements – should therefore explicitly address uncertainty or risk losing this critical aspect of students' experience. In this paper we present a framework for teaching students to assess and communicate their uncertainty, which is grounded in best expert practices for conveying uncertainty and familiar terms-of-art in geology. The starting point of our framework is the recognition of uncertainty in both geologic data and models, the latter of which we use as an encompassing term to refer to potential geological processes and structures inferred on the basis of incomplete information. We present a concrete application of the framework to geological mapping and discuss how it could enhance student learning in both traditional in-person and virtual experiences. Our framework is extensible in that it can be applied to a variety of geologic features beyond those where uncertainty is traditionally assessed, and can also be applied to geological subdisciplines.

## 1 Introduction

Capitalizing fully on scientific research requires understanding how much uncertainty surrounds it (Fischhoff and Davis, 2014; Kirch, 2012). In many cases these uncertainties extend beyond those associated with the measurement of observable phenomena and objects. This is particularly true in geoscience, where high levels of uncertainty are the standard, rather than the special case (Bárdossy and Fodor, 2001; Frodeman, 1995). For example, most field-based geoscientists are familiar with the limitations on observation imposed by incomplete exposure. Even in cases where exposure is exceptional or complete, geometric and lithologic complexities manifested over a wide range of spatial scales may lead to distinctly different interpretations by individual scientists. In this way, learning to cope with uncertainty in its many forms is a central component in the training of an expert geoscientist.



Exactly how this occurs – how the characterization, assessment, and conveying of uncertainty changes from the novice to expert mind – is not well understood (Petcovic et al., 2009). In-part, this is because sources and types of uncertainty change depending on the scientific question and methodology, and so learning to cope with uncertainty is inherently specialized

(Fischhoff and Davis, 2014). But it is also because the subjective inferences geoscientists make when faced with uncertainty have generally been an *implicit* element of the data collection and interpretation process (Bond, 2015). It is perhaps therefore unsurprising that formal geoscience curriculum never (or very rarely) explicitly addresses uncertainty and best practices for handling it. We posit that, for many students, field mapping is the first exposure to the underlying uncertainties inherent in geologic data and processes. For example, Petcovic et al. (2009) report the following anecdote from a novice mapper:

40       *"…it was hard for me to identify [rocks] in the field because of the Ward's samples that were given in the lab, which*
      *are perfect representations of perfect rocks and minerals. I can honestly say that absolutely none of them … come*
      *with algae and weathering and erosion and anything else subject to the environment…[note: Ward's Scientific is a*
      *company that commonly supplies samples for education]"*

This anecdote exemplifies the common experience of geology students wrestling with uncertainty in the field. Geology field educators will recognize that, while some students can be overwhelmed by uncertainty in observations, others can systematically underestimate uncertainty in their preoccupation with determining the "right answer" to a problem or field area, rather than attempting to establish a plausible interpretation that is most consistent with the data and their relative uncertainties. In both cases, what students need is language to understand the nature of uncertainty in geology – a

combination of uncertainty due to imperfect information in the world, and uncertainty due to the human mind that observes the information. This need has only become more timely and relevant with the move to replace traditional field mapping with virtual experiences, brought on by the COVID-19 pandemic. Virtual field activities that rely on pictures, maps, and previously collected measurements run the risk of obscuring uncertainties that would otherwise have a prominent influence on scientific interpretation (if experienced in-person). How reliably can one infer the 3D structure of an outcrop from a series

of 2D images? How certain can one be about the location of a lithologic boundary from satellite imagery alone?

Here, we address this pressing need by presenting a framework for teaching students to characterize, assess, and convey uncertainty that is suitable for both traditional in-person and virtual field experiences. The starting point of our framework is recognizing that there is uncertainty in both geologic data and models of geologic processes. We use the term data to refer

to any geologic information (quantitative or qualitative) relevant to scientific judgment. We use the term model as an encompassing term to refer to potential geological processes and structures inferred on the basis of incomplete data: it includes model as it is traditionally used, as an account of what happened in the past to yield the current state of the Earth, as well as the related terms of theory, hypotheses, inference, and conjecture.  The reader might object to having all of these





terms treated as the same, but we are not arguing that they mean the same thing, only that they all refer to speculations
about the Earth that are drawn out of observations. Further, the reader may note that an important distinction among the
terms is the degree of evidence in support of the speculation (i.e. a simple inference is likely supported by less positive data
than is a widely accepted theory).  This distinction is in part what we seek to make explicit for students and experts in the
form of uncertainty.

The difference between data and model uncertainty is a useful high level categorization that is applicable to other sciences,
and conceptually similar to other scientific uncertainty categorization schemes in the learning sciences literature (Costanza
and Cromwell, 1992; Pickering, 1995; Metz, 2004; Manz, 2015). However, what are the "data", and the resulting models, will
differ widely across sciences and even within subdisciplines of geology. In the next section, we present a detailed uncertainty
rating scale suitable for geologic mapping in traditional and virtual field exercises. An important feature of our rating scale is
that it allows for assessment of uncertainty in qualitative observations, which makes up a large portion of geologic data. To
the extent that students are familiar with assessing uncertainty in science, they are likely to be exposed to the more statistical
approach of taking multiple quantitative measurements and estimating uncertainty based on numerical variability (Bárdossy
and Fodor, 2001). Yet, under many field conditions this approach is simply not feasible due to resource and time constraints,
or impossible due to the qualitative nature of the observation.

**2 Uncertainty rating scale**

Our work builds on existing best practices for assessing and conveying uncertainty on geologic maps, where uncertainties in
the location of faults and contacts between rock types are notated using solid lines (denoting high certainty), dashed lines,
or dotted lines (denoting high *un*certainty; see Fig 1A). This blunt categorization scheme used for mapping offers a useful
starting point for thinking about how geoscientists codify their uncertainties in the underlying observations and models.
However, that system has significant limitations, the first of which being its applicability to a relatively narrow range of
relevant observations (e.g. map-linear contacts). There is also no standard for how to use the low and high uncertainty
categories except by implicit inference from community practice. Finally, the dynamic range is likely compressed by experts
being unwilling to publish high uncertainty observations or inferences, and having only three categories does not allow the
community to distinguish gradations of uncertainty, which is likely to be particularly important for students.

To address these limitations, we developed an explicit and extendable six-category ranking system ("uncertainty rating
scale") for conveying the uncertainty associated with observations and models, grounded in familiar terms of art in geology
(see Fig. 1B). The categories are: **No evidence**, **Permissive**, **Suggestive**, **Presumptive**, **Compelling**, and **Certain**. "No evidence"
and "Certain" are end members, because there is no variability within these categories. No evidence indicates there is no



information that constrains an observation or model in any way. Certain indicates that an observation is unambiguously

accurate. We note that one key difference in the uncertainty rating scale as applied to observations when compared with

models is that models cannot be regarded as certain. The middle four categories – Permissive, Suggestive, Presumptive,

Compelling – have a range of possible values. Permissive is the least certain form of evidence.  Permissive suggests that a

particular observation or model cannot be ruled out, but it is also not the only available solution.  Suggestive indicates that

there is positive evidence for a particular observation or inference, but that the evidence also allows the possibility for other

inferences.  Presumptive – presumed in the absence of further information – indicates that an observation or model is more

likely right than wrong. Compelling indicates that the evidence is strongly supportive of the observation or model, and is

necessarily based on a preponderance of positive evidence.

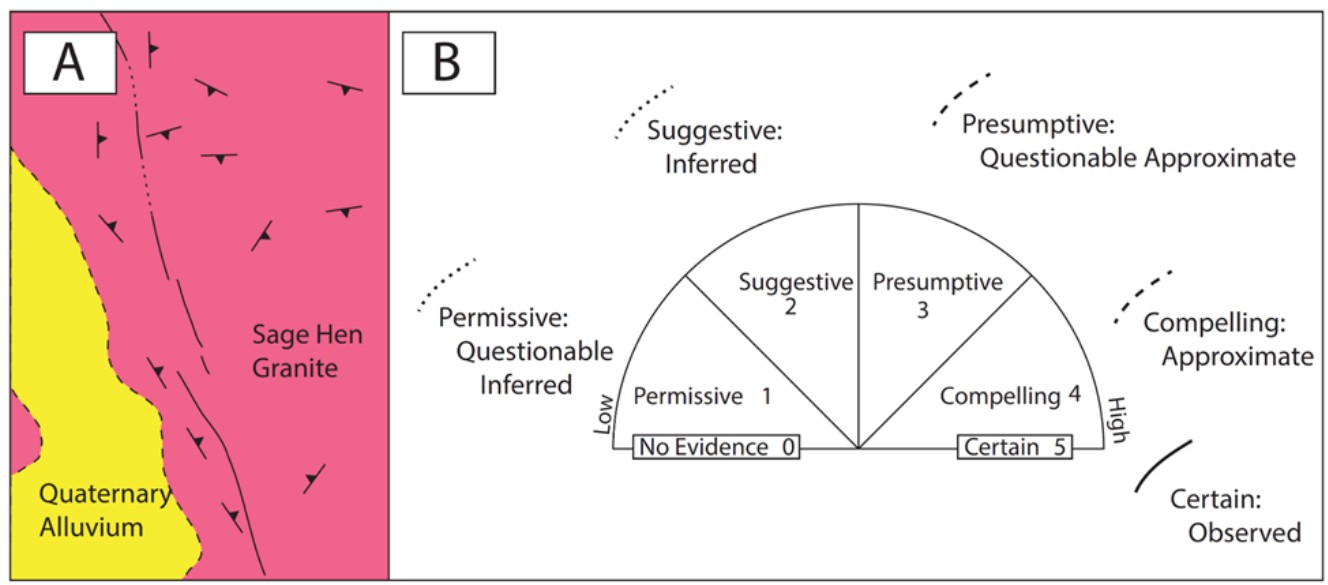


**Figure 1:** (A) Fragment of a geologic map of the Mount Barcroft-Blanco Mountain Area of Eastern California by Ernst and Hall (1987) showing moderate uncertainty in contact between older alluvial deposits (yellow) and Sage Hen Quartz Monzonite (pink), represented by a dashed line. The map also shows a fault in the Sage Hen unit, represented by a solid line where the
fault is directly observable, and a dotted line when uncertain. (B) Evidence meter that captures a larger dynamic range in uncertainty than the three-fold categorization used in A.

The uncertainty rating scale is extensible in that it can be applied to observational data about geologic features beyond those

where uncertainty is traditionally assessed (e.g. map-linear contacts). Through interdisciplinary fieldwork with geologists and

cognitive scientists, we identified four critical properties of an outcrop that could vary in uncertainty: **Attachedness**,

**Lithological Correlation**, **3D Geometry, and Kinematics**. Attachedness is the determination of whether the rock exposed at

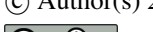



the Earth's surface is directly connected to the rocks below the surface at that location; for example, a geologist who sees a small rock in a field but can not see the base of the rock might think, "well it could be attached to bedrock," might record the observation as *permissible*. Lithological Correlation is the determination of whether a particular rock belongs to a known

formation; for example, a geologist may observe some unique structures or fabrics locally within formation A at its type locality, and subsequently conclude that those same structures or fabrics are *suggestive* of the presence of formation A at a new, uncategorized location. 3D Geometry is the determination of the internal features of an outcrop (such as the orientation of bedding or other fabric) or the contact(s) between distinct features/units/lithologies in a field area; for example, when geologists measure the strike and dip of bedding at an outcrop, it is generally considered *presumptive* that the orientation is

representative of that present in the subsurface, at least for some reasonable distance around where the measurement was taken. Kinematics is the determination of the past movement of the rock, such as along a fault; for example, most geologists would take the repetition of a diagnostic sequence of well-exposed stratigraphy across a mapped fault as *compelling* evidence of shear sense.

The uncertainty rating scale was developed for simple observations and models/inferences, the type students are most likely to focus on during field mapping, e.g., unit identification, the location of a fault or contact that isn't visible in outcrops, sense of movement across a fault, etc. Using this example, a *presumptive* model would be one that is more likely correct than not by virtue of being consistent with some presumptive data (or a lot of suggestive data), and inconsistent with only some suggestive or permissive data, where there is no other model that accounts for the same or more data. However, the rating

scale could readily apply to much larger models or interpretations about how the Earth works, such as unresolved alternative hypotheses for a region, e.g., flat slab versus hit-and-run mechanism for Laramide deformation in North America (Maxson and Tikoff, 1996; English and Johnston, 2004), gravitational sinking versus proto-subduction for the formation of early continents (Brown et al., 2020).

**3 How might the uncertainty rating scale enhance student learning in traditional and virtual field experiences?**

Broadly, there are two means through which the new uncertainty rating scale could benefit students during virtual and in-person field experiences: one is through the actual **practice** of explicitly characterizing, assessing, and conveying uncertainty, and the other is through the **consumption** of science where uncertainty is (or is not) communicated. For both, there is theoretical evidence uncertainty might benefit students (novices) and experts alike (Larrick and Feiler, 2015). However, here

we focus on the particular advantages for students.



In field practice, the presented geologic mapping framework provides students with a language to explicitly communicate uncertainty that they already feel, and that is true of the world. In science, it is often necessary to admit provisionally some assumptions that are not well-established, and this is particularly true for students who are working with less disciplinary
knowledge than experts. Having a framework for conveying their uncertainty could help relieve students from some of the psychological distress that is characteristic of "not knowing", or affirm the development of their scientific ability by making progress (in the form of reduced uncertainty over time) more conspicuous. Our own experience with students in the field suggests that they are more likely to focus on their uncertainty than ignore it, but for some, an explicit framework for communicating uncertainty would enforce remembering what is only supposition – preventing the forgetting of uncertainty.
There is a tendency of the mind to "rest on an assumption" when it appears to fit in with other knowledge, and to forget that it has not been proved (Beveridge, 1957; Cantor, 1991). Students (and experts) who do not record uncertainty, therefore, risk a rose-colored glasses effect whereby past uncertainties are systematically underestimated to achieve concordance or consistency.

All the potential benefits of the uncertainty framework for geologic mapping in the field are also relevant to virtual practice. But unlike in the field, where students are consistently confronted with uncertainty, virtual activities may obscure uncertainties that would be otherwise prominent; the worry is not that students will forget about initial uncertainties, *but that they will not recognize them in the first place*. According to the cognitive science literature, structured workflows could help scientists to recognize uncertainties they might otherwise overlook by explicitly enforcing the consideration of ideas
that would not have been fully deliberated otherwise (Soll et al., 2016; Wilson et al., 2019). For example, Macrae et al. (2016) asked expert geoscientists to interpret a 2-D seismic reflection image and found that although experts reported that they were considering geological history when making interpretations using their familiar workflows, only experts in a structured workflow group showed evidence of effectively considering geological evolution of their interpretation. Extrapolating to the uncertainty framework, this suggests that students (and experts) might feel they have appropriately considered uncertainty,
but unless uncertainty assessment is made explicit through a workflow, there is a risk of their mismanaging and inappropriately weighing uncertainty.

When it comes to consuming science, having access to the uncertainties associated with the practitioners' data and models can help students' fill-in missing context. This is perhaps most critical for consumption of science in virtual field experiences,
where uncertainty judgments about attachedness, lithological correlation, 3D geometry, or kinematics can reveal features of an environment that otherwise could not be perceived from available imagery.  Most importantly, however, having access to others' uncertainty will make explicit that *disagreements in uncertainty exist*, and that these disagreements can be resolved in different ways to yield different geologic predictions. We believe that this will promote a more sophisticated




discourse amongst student teams and support the consideration of multiple working hypotheses. In the next section, we

present an initial framework for describing the relationship between data and model uncertainty to students.

## 4 Uncertainty in data and the models they produce

Students, by virtue of having less knowledge and experience than experts, will likely experience greater uncertainty in making observations and collecting data. Students' uncertainty in data will necessarily feed forward into their model uncertainty, as shown in Figure 2, because a model is only as good (certain) as the data that informs it. Inferences based on uncertain data

may increase the likelihood of bias in model selection – in the absence of certain data, a scientist is more likely to rely on whatever model is most accessible in their mind, perhaps what they worked on most frequently, most recently, or what is most salient. This tendency to default to the interpretation that is most dominant in mind is referred to as the "availability heuristic" (Tversky and Kahneman, 1973), and it can lead to bias in experts (Bond et al., 2007) and students (Alcalde et al., 2017). For example, Alcalde et al. (2017) found that geology students were more likely to interpret a fault in a seismic image

as normal and planar because this geometry was overrepresented in teaching materials the students encountered; after students were exposed to a greater range of geometries through training, the range of interpretations increased. In short, one rationale for recording and communicating uncertainty is that doing so can reduce bias (Macrae et al., 2016).

The relationship between data and model uncertainty is also bi-directional, in that model uncertainty also feeds backward

into data uncertainty (see Fig 2). Predictions based on uncertain (potentially biased) models guide what new data to collect, but can also lead to re-evaluation of previously collected data and their related uncertainty. Depending on how accurate the selected model is, either at predicting new data or accounting for previous data, the certainty of the model will also change. This conception is, in practice, why the concept of multiple working hypotheses is so important within science (Chamberlin, 1890). If left in isolation, a single preferred hypothesis/model is more likely to serve as its own reinforcement than it is to

provide a basis for future discoveries.

## 5 The way forward

In this paper we have provided a workflow for recording and communicating uncertainty for basic mapping. We have grounded the effort in principles of geology and the role of the mind in the practice of science. The nature of uncertainty

associated with primary observations relevant to the subdomains of geoscience are likely broad and best described by experts. Developing a system for those specific observations will likely require a community effort. Our informal impression from discussion with sub-domain experts is that the existence of uncertainty and the value in capturing it is immediately



recognized, but exactly how that should be done and for what properties is not always clear. Thus, in this section we briefly review our process for coming to the properties reported, in the hope that others can use it to develop a workflow by analogy.


**Figure 2:** Bi-directional influence of data and model uncertainty.

Our basic principles, which we suspect will be broadly applicable, are 1) uncertainty in any observation can be categorized into a scale that ranges from completely uncertain to completely certain, and 2) uncertainty of different properties should
be orthogonal as much as possible, i.e., uncertainty in one property should not correlate with another. Wherever uncertainty is highly correlated across properties, they are not independent and thus are candidates for being collapsed into a single property. We applied these principles in the field while mapping the Sage Hen pluton and surrounding rocks in the White-Inyo mountains of California. We choose this area specifically because there exist two different published maps that differ



markedly in their interpretations of the area, reasoning that if experts had come to significantly different conclusions from
similar observations it was likely the underlying data had significant levels of uncertainty. As we mapped at each stop we
asked, "what properties would be important to record here, and how certain are we that what we have recorded reflects
the true state of the world?" After the first day of mapping we asked, "are there any properties we have not recorded but
would record if we found evidence for?" These questions relatively quickly established the basic properties: lithology ("what
are the rocks?"), attachedness ("are these rocks in-place?"), 3D geometry ("how are the rocks or contacts oriented?"), and
kinematics ("do the rocks show evidence of movement?").  The subsequent challenge was grounding the categories within
each property.  A discussion at an outcrop was often required to come to agreement about a category.  Experts began by
privately assessing uncertainty and then sharing to see how well they agreed and to discuss the origin of any disagreements,
if there were any.  We observed that a useful touchstone for coming to consensus was the mid-point.  Here experts could
generally agree about whether a given observation allowed one to determine if a property was more or less likely than all
other possible interpretations.

We recognize two unresolved questions from our work. The first is whether some categories should be interpreted as
reflecting a quantitative range of spatial uncertainty.  For example, perhaps mapping a contact as suggestive should imply
that there was a spatial range for its location (e.g., +-100 meters)? The second is what the best way would be to establish
the scales so that the community can use them in the same way. During our small-group trip, we mutually adjusted to using
the scale similarly after a few days such that there was high agreement on independent judgments of uncertainty. Likely a
lifetime of experiences with the range of outcrop quality offered a common ground to scale uncertainty. The role of
experience in grounding the scale highlights the role of the academy in developing a profession that can adapt this framework
for capturing and communicating uncertainty.


We might also ask how should students be taught to use an uncertainty scale in a way that maximizes the consistency of the
use of the scale and thus consistent application of the scale across users?  We propose an approach that has two parts,
recognizing that there is not a purely objective uncertainty for any given observation.  Rather, the uncertainty is a
combination of the skill and background of the observer and the rocks they are presented with. To educate students about
the uncertainty that can be attributed to the rocks we employed a strategy of showing pictures of outcrops accompanied by
an expert's rating of uncertainty and their reasoning for that rating (these are available in the materials described in section
7). Further, students would need practice introspecting about how uncertain they feel about the decisions they are making.
By developing a practical pedagogy that can tune observers' use of the scale, the community will have a mechanism for
aggregating trustworthy data.






## 6 Conclusions

In this contribution, we address the current gap in undergraduate geoscience education on uncertainty. We present a framework for teaching students to characterize, assess, and convey uncertainty that is suitable for both traditional in-person and virtual field experiences. The 6-fold categories for data uncertainty are: No evidence, suggestive, presumptive, compelling and certain. For model uncertainty, the same ranking applies except that there are no "no evidence" and "certain" characterizations. We hope geoscience educators find the framework relevant and useful; to aid in its adoption, we have provided materials for teaching about data and model uncertainty (described below). The materials were designed for basic geologic mapping education, but can be easily adapted, as discussed in the previous section.

The benefits of the uncertainty framework for students is that it: (i) provides relief from the distress of "not knowing" by providing a language to communicate uncertainty; (ii) affirms scientific growth by making reductions in uncertainty more conspicuous; (iii) prevents systematic underestimation of uncertainty in field practice, and a failure to recognize uncertainty in virtual practice, by enforcing explicit consideration; (iv) allows access to others' uncertainty fills-in missing context that otherwise could not be perceived in virtual activities; and (v) makes the nuances of geologic interpretation more apparent by demonstrating that disagreements in uncertainty exist, and can be resolved in different ways, to produce different geologic predictions. In addition to these, we posit there are longer-term benefits from exposure to the uncertainty framework that will continue to serve students after the virtual or in-person field experience has ended. For students pursuing careers in geology, for whom managing uncertainty will be fundamental, the framework can provide a critical scaffold on which new uncertainties can be built. Yet, even for those students who do not continue in science (which is nearly half; National Science Board, 2018; Chen, 2013), the uncertainty framework is still beneficial because it has the potential to move students from a black-and-white worldview (e.g., an interpretation is either right or wrong) to one of more nuance, where data has varying quality and interpretations have different strengths and weaknesses. We believe that this is the essential first step toward acceptance that published work is not "fact", even if published by scientists with strong reputations. This kind of appreciation of the presence of uncertainty in science is a key insight for calibrating public trust in science (Steijaert et al., 2020).

## 7 Materials for teaching uncertainty in geologic mapping

We provide materials for teaching about data and model uncertainty in the form of a virtual field-camp / capstone activity exercise developed (in part) by the authors. This exercise includes lecture and reading material, Google Earth referenced data files and images, and student assignments structure on a day-by-day basis. The module is intended to be conducted



over a period of 5-6 days. All materials are freely available from the Science Education Resource Center (https://serc.carleton.edu/NAGTWorkshops/online_field/activities/238026.html).

**8 Code availability**

Not applicable

**9 Data availability**

Not applicable

**10 Executable research compendium**

Not applicable

**11 Sample availability**

Not applicable

**12 Video supplement**

Not applicable

**13 Supplement link**

https://serc.carleton.edu/NAGTWorkshops/online_field/activities/238026.html


**13 Team list**

Not applicable



## 15 Author contributions

CGW, RTW, TFS, and BT developed the uncertainty rating scale and associated categorizations during field outings. CGW prepared the manuscript with contributions from all co-authors

## 16 Competing interests

The authors declare that they have no conflict of interest


## 17 Disclaimer

This research has been supported by the National Science Foundation, Future of Work at the Human Technology Frontier (grant no. 1839705)

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
