# Peer review of "Teaching Uncertainty: A new framework for communicating unknowns in traditional and virtual field experiences"

_Solid Earth, 2021_

## Referee Comment (RC1)

**Teaching Uncertainty: A new framework for communicating unknowns in traditional and virtual field experiences**

Cristina G. Wilson[1,2], Randolph T. Williams[3], Kathryn Bateman[2], Basil Tikoff[3], Thomas F. Shipley[2]

I was excited to read this paper, but whilst agreeing with the over-arching conclusions I felt that it was lacking in several places.

I detail my main thoughts below and have also attached a commented copy of the manuscript.

Firstly, I felt that there was an overplay on the virtual aspect. All the concepts and issues identified also apply to traditional field experiences. I felt like the virtual 'hook' had been added to make it relevant to the special publication. I would not differentiate between in person and virtual field experiences. If you do differentiate virtual field trips from physical in-field experiences I think you need to back-up any assertions with evidence, you don't do this. I have anecdotal evidence from this year that contradicts your premise - that uncertainty is not appreciated in a virtual field trip. So, it would be really nice to know if you have anecdotal and preferably non-anecdotal evidence to back-up your assertions. Also, when you go into your example of using the categorisation it seems that this was actually done physically in the field.

Secondly, you need to give background on how geology is taught e.g. pre-field practical exercises as well as in field-practice. From my experience uncertainty and interpretation is inherently a part of geoscience teaching from day 1. Drawing cross-sections from maps, mapping with solid, dashed and dotted lines, green-line mapping, shading outcrops etc… are all explicit references to data and certainty. At the moment the paper implies it is not taught at all and a students' first experience/thought of uncertainty is in the field.

A broader discussion of current practice beyond the solid, dashed and dotted lines I believe is essential for context. You describe the current depiction of boundary lines as blunt. But I question if it is 'blunt', or if it is fit for purpose? Qualitative scales are very hard to define and are very subjective – you do not really draw this out, or the literature on this topic; which seems essential. Could a six-categorisation set of certainties be effective?

I like the idea of broader application beyond boundaries and it would be really useful to have a set of examples for each application (Boundary, Attachedness, Lithological Correlation, 3D Geometry, and Kinematics) for each category (No evidence, Permissive, Suggestive, Presumptive, Compelling, Certain). I think this would really help get the reader on-board. I struggled to think of instances when you would use No evidence – is this really a category? And scenarios in which say for 'attachedness' I would choose presumptive over suggestive, over permissive. For me it is important to note that mapping is an iterative exercise (see discussion in Jones et al. 2004) and the field map has an associated field notebook which provides the metadata – the thoughts, ideas, hypothesise and uncertainties. So, for example, if I mapped a rock as a unit, but thought it might not be attached e.g. it may be drift. I would then proceed to see if it was lithologically distinct from the surrounding rock units, then compare any structural readings to surrounding outcrops etc… build-up evidence and make a decision. This metadata seems to be crucial, but is not really discussed or described. This is also what I would teach students in the field if they were uncertain about attachedness. It

might also be worth reviewing the difference between a field map and fair copy map. The two have quite different jobs.

The section on applying the categorisation is a bit disappointing – you note that you tried it in the field with experts. But there is no data on what the different experts found. How they reached a consensus, how far apart they were in their categorisations. Did it work, was it practicable? Then you move on to discuss trialling with students – but again there is no data on this to consider, to see if it works. You say you showed them photographs of outcrops with an experts rating and reasoning of certainty. But what did they learn… could they apply that themselves? I looked at the NAGT link and tried to access the student questionnaires, but I can't see the questions for students without filling in the form.

For me the paper really needs more background on how mapping is taught, uncertainties recorded and how these are then iterated into models. But perhaps more crucially it needs the data on how effective the categorisations are. Can they be practically used? What were the students and experts thoughts on the categories, and their ability to apply them?

I am happy to chat to you about any of the comments I have made if that would be useful. I think you make some really valid points about the explicit teaching of uncertainty and I agree with your over-arching conclusions, but the evidence for me is not there for your categorisation approach and application. I have recommended rejection, as I feel any new submission would be quite a different paper.

Best wishes
Clare

[revised manuscript text omitted]